# Identifying Intraoperative Spinal Cord Injury Location from Somatosensory Evoked Potentials’ Time-Frequency Components

**DOI:** 10.3390/bioengineering10060707

**Published:** 2023-06-11

**Authors:** Hanlei Li, Songkun Gao, Rong Li, Hongyan Cui, Wei Huang, Yongcan Huang, Yong Hu

**Affiliations:** 1Institute of Biomedical Engineering, Chinese Academy of Medical Sciences and Peking Union Medical College, Tianjin 300192, China; 2Department of Rehabilitation, The 2nd Affiliated Hospital of Guangdong Medical University, Zhanjiang 524255, China; 3Shenzhen Engineering Laboratory of Orthopaedic Regenerative Technologies, Orthopaedic Research Center, Peking University Shenzhen Hospital, Shenzhen 518036, China; 4Department of Orthopedics and Traumatology, The University of Hong Kong, Hong Kong SAR, China

**Keywords:** machine learning, naive Bayes, somatosensory evoked potentials, spinal cord injury, time-frequency components

## Abstract

Excessive distraction in corrective spine surgery can lead to iatrogenic distraction spinal cord injury. Diagnosis of the location of the spinal cord injury helps in early removal of the injury source. The time-frequency components of the somatosensory evoked potential have been reported to provide information on the location of spinal cord injury, but most studies have focused on contusion injuries of the cervical spine. In this study, we established 19 rat models of distraction spinal cord injury at different levels and collected the somatosensory evoked potentials of the hindlimb and extracted their time-frequency components. Subsequently, we used k-medoid clustering and naive Bayes to classify spinal cord injury at the C5 and C6 level, as well as spinal cord injury at the cervical, thoracic, and lumbar spine, respectively. The results showed that there was a significant delay in the latency of the time-frequency components distributed between 15 and 30 ms and 50 and 150 Hz in all spinal cord injury groups. The overall classification accuracy was 88.28% and 84.87%. The results demonstrate that the k-medoid clustering and naive Bayes methods are capable of extracting the time-frequency component information depending on the spinal cord injury location and suggest that the somatosensory evoked potential has the potential to diagnose the location of a spinal cord injury.

## 1. Introduction

Spinal cord injury (SCI) remains the most worrisome complication of corrective scoliosis surgery [1,2] and can even lead to paraplegia in severe cases. Since surgery for scoliosis usually involves multilevel distraction and fusion of the thoracic and lumbar vertebrae, distraction is an important mechanism for SCI in corrective scoliosis surgery [3]. According to the guidelines published by the American Clinical Neurophysiology Society, when SCI occurs, surgeons should look for any mechanical damage, reducing the degree of distraction, adjusting retractors, removing or adjusting grafts or hardware, and prompting the anesthesiologist to quickly raise the blood pressure [4,5]. If the source of injury can be removed promptly, spinal cord function can still be restored. Thus, accurate diagnosis of the SCI location during corrective scoliosis surgery will help reduce the time the surgeon needs to investigate.

Techniques for intraoperative imaging, which are an important auxiliary means for spinal surgery, are constantly developing [6]. However, there are still some shortcomings in the application of intraoperative image-guided technology, such as high cost, radiation exposure, long image acquisition time, and unstable image quality [6,7,8]. In recent years, the use of intraoperative neurophysiological monitoring (IONM) has increased in order to avoid neurological complications, and somatosensory evoked potential (SEP) is the most commonly used IONM method [9]. The SEP consists of cortical responses generated by peripheral stimulation electrodes. It can monitor perioperative neurological changes in the sensory pathway [9,10]. Detecting the decreasing amplitude or prolonging latency of the SEP provides early warning of possible damage to the sensory pathway. In addition, the SEP has been reported to support the precise localization of an SCI and to diagnose the cervical level of damage in cervical myelopathy [11,12]. Unlike traditional measurements using amplitude and latency, SCI location detection is accomplished by identifying changes in the time-frequency component distribution (TFD) of the SEP [13,14]. Different components of the SEP have various origins along the somatosensory pathway [11]; therefore, different locations of SCI result in different distribution patterns of the time-frequency components (TFCs) of the SEP. However, existing studies have only demonstrated the predictive ability of the SEP for the location of SCIs in the cervical spine. Therefore, in the present study, we investigated whether the SEP could be used for the identification of SCI at other locations of the spine.

In order to achieve SCI location identification, it is necessary to utilize the information contained in the TFD of the SEP as much as possible. A change in SCI location causes SEP TFCs to have different distribution boundaries. In addition, the number of TFCs in many subspaces of the time-frequency space is also affected by the change in SCI location. Previous studies have chosen support vector machines (SVMs) to extract the distribution boundaries of SEP TFCs to achieve the goal of identifying SCI locations [11,15]. However, this method only uses the distribution boundary information of the SEP TFCs. As reported, the SVM is susceptible to outlier interference [16]. The quantitative or distribution probability information of TFCs can suppress the influence of random noise. A study applied the random forest to the multiclass classification of SCI locations by repeated random sampling to construct multiple decision trees, each of which can classify SCI locations with the boundary and quantitative information provided by some of the training data [12]. When revealing the importance of the TFCs of a subspace, statistical analysis of randomly sampled sample points needs to be carried out in combination with the classification results, and the corresponding methods have not been outlined in previous studies. The k-means clustering algorithm has been used to reveal the stable distribution area of the TFCs of a normal SEP [17]. If SEP data from different locations of SCI are included in training, the clustering models would be more widely applicable, and the correspondence of TFCs with different SCI conditions can thus be established.

In order to detect the location of the SCI intraoperatively to perform remedial action as soon as possible, we explored the correlation between the SEP TFD and the SCI location and developed an SCI location identification method based on SEP TFCs. Firstly, in order to obtain high-resolution TFCs, the SEP was decomposed by matching pursuit (MP), and all the TFCs were described in terms of latency, frequency, and energy. Secondly, to realize the partitioning of the time-frequency space, the distance-based clustering algorithm was used to cluster the TFCs. To emphasize the differences in the number of different SCIs in each TFC cluster, we increased the number of centroids, that is, the time-frequency space was partitioned into smaller subspaces. The combination of multiple different time-frequency subspaces can form an arbitrary SCI TFD pattern. The number of TFCs of different SCIs in each subspace was also counted. The clusters of TFCs, i.e., time-frequency subspaces, were treated as features, and the number of TFCs of different SCIs in that subspace was the feature value. Through feature selection, the distribution boundaries of the TFCs can be extracted and utilized, as well as the quantity information of TFCs in each subspace. Finally, a naive Bayes classifier was constructed, which uses the quantity information of TFCs for SCI location identification.

The TFD of the SEP in rats has been reported to be similar to that of humans [18,19]. We previously developed an experimental rat model to simulate intraoperative distraction SCI by applying distraction to the spine [20]. We recorded the SEP signals in the experimental rat model after applying distraction SCI at 19 spinal levels, respectively. Using this model, we constructed a naive Bayes classifier, which maintained similar accuracy to previous studies in the identification of cervical SCI locations and achieved the goal of classifying SCI locations in the cervical, thoracic, and lumbar spine. Therefore, the SEP has the potential to identify SCI locations not only at the cervical spine but also across broader spinal ranges. We provide an effective intraoperative SCI localization scheme that can improve diagnostic efficiency. This study also explores the effect of SCI location on the SEP TFD, and it will help to determine the origin of specific SEP TFCs.

## 2. Materials and Methods

In this section, animal model construction, data collection, feature extraction and selection, and classifier construction are stated. The data processing flow is shown in Figure 1.

### 2.1. Animal Model and Data Collection

As shown in Table 1, 210 female Sprague-Dawley rats (specific-pathogen-free level, aged 7 to 8 weeks, weight 280 to 320 g) were purchased from the Guangdong Medical Laboratory Animal Center (license No. SCXK (Yue) 2018-0002) and randomly assigned to 1 normal group (*n* = 20) and 19 SCI groups (*n* = 10). The 19 groups of rats were assigned to the cervical injury group, the thoracic injury group, and the lumbar injury group, including 2, 11, and 6 groups of rats, respectively. The normal group only received anesthesia and SEP collection. The cervical group was injured at C5 and C6; the thoracic group was injured at T1–T4 and T7–T13; and the lumbar group was injured at L1–L6.

In a previous study, a distraction SCI was produced in rats using customized vertebral clamps [20]. For example, the distraction injury between cervical vertebra 5 and 6 (C5/C6) was denoted as C5, and the procedure was as follows: Animals were anesthetized for SCI and evoked potential testing and sacrificed using intraperitoneally injected pentobarbital sodium (60 mg/kg; Sigma, St. Louis, MO, USA) and xylazine (10 mg/kg; Sigma). The rats were placed on a thermostatic pad at 37 °C to receive a subcutaneous injection of 5 mL physiological saline solution to prevent dehydration. Using standard aseptic principles and techniques, dorsal ligament resection and facet arthrotomy were performed at the C4–C7 interspace. Customized vertebral clamps were used to rigidly hold the transverse processes of C4–C5 and C6–C7. The respective clamps of C6–C7 were distracted rostrally and caudally to produce a displacement of 3 mm and held for 1 s before being returned to their initial position. Figure 2 shows the recording of the surgical procedure.

Immediately after the SCI, electrophysiological evaluation (YRKJ-G2008; Yirui Technology Co., Ltd., Zhuhai, China) was conducted. Tibial SEPs were evoked from stimulation proximal to the ankle via a pair of needle electrodes (NE-S-1500/13/0.4; Friendship Medical Electronics Co., Ltd., Xi’an, China) using the following parameters: 0.1 ms duration, 5.3 Hz frequency, and 3–5 mA intensity (to elicit mild toe twitching). Recordings were collected using two scalp needle electrodes subcutaneously inserted over the primary somatosensory cortex and a frontal midline reference electrode. The recorded signal was amplified 2000 times at sampling rate of 10 kHz with a bandpass filter between 30 and 3000 Hz. The SEP signals were averaged over 200 responses for each trial [11,12]. In this study, all signal processing routines used for the analysis were developed in MATLAB (version R2019a; MathWorks, Natick, MA, USA).

### 2.2. Time-Frequency Analysis

In this study, obtaining reliable TFCs is the basis for subsequent feature extraction and classification. Therefore, the high-resolution MP algorithm was used for time-frequency decomposition. The MP algorithm represents the SEP signal as a linear combination of TFCs:(1)$$S(t)=\sum_{m=1}^{M}g_{m}(t)+e(t),$$
where *g_m_*(*t*) represents TFCs after decomposition, and *e*(*t*) represents the residual component. In the MP algorithm, TFCs are selected from redundant dictionaries, and the Gabor dictionary has been recommended in previous studies. Therefore, the m-th TFC can be described as follows:(2)$$g_{m}(t)=ae^{-\pi {\left[(t-T)/\sigma \right]}^{2}}\cos(2\pi f(t-\tau )+\phi ),$$
where *T*, *f*, *a*, *σ*, and *ϕ* define the latency, frequency, amplitude, span, and phase of *g_m_*(*t*), respectively. The TFC is generally selected by an iterative algorithm. In the initial step, a TFC is analyzed by identifying the waveform *g*_1_(*t*) with the highest inner product with the signal *S*(*t*). At the same time, we obtain the residual, which is the difference between *S*(*t*) and *g*_1_(*t*). Then, *S*(*t*) is iterated with residuals and the process is repeated to determine a new *g_m_*(*t*), until the total energy of the TFCs reaches 99.5% of *S*(*t*). During iteration, the values of *T*, *F*, *a*, *σ*, and *ϕ* are constants. The relative energy $\bar{E_{m}}$ of a TFC *g_m_*(*t*) is calculated as follows:(3)$$E_{m}={\sum_{t}\left|g_{m}(t)\right|}^{2}.$$(4)$$\bar{E_{m}}=E_{m}/\sum_{t}{\left|S(t)\right|}^{2}.$$

Finally, *T*, *f*, and $\bar{E_{m}}$ were selected to describe all the TFCs. For the details of the MP algorithm, please refer to [17,21,22,23].

### 2.3. Clustering of Time-Frequency Components

In order to explore the local characteristics of the TFCs, we used the k-medoids clustering method to divide the hindlimb SEP TFCs of all rats into multiple component clusters. Different from k-means, which uses the mean value of objects in the cluster as the center in the iterative process of searching for the optimal centers, the k-medoids algorithm selects the object with the minimum Euclidean distance in each cluster. The k-medoids algorithm can deal with outliers better than the k-means can [24]. The silhouette coefficient was used to evaluate the effect of the number of clustering centers on the clustering results [25]. For the i-th object, the silhouette coefficient is calculated as follows:(5)$$S(i)=\frac{b(i)-a(i)}{\max\{a(i),b(i)\}},$$
where *a*(*i*) is the mean distance between the i-th vector and all the other points in the cluster that it belongs to, and *b*(*i*) is the mean distance between the i-th vector and all points in the nearest cluster. The silhouette coefficient ranges from −1 to 1. The larger the value, the better the clustering performance. The TFC clusters are referred to herein as features. For each SEP, the features with the presence of TFCs were denoted as 1, and those with no TFCs were denoted as 0.

TFC clusters obtained by k-means clustering were also taken as features to compare the classification effects.

### 2.4. Feature Selection

To remove random noise components and select the most valuable features for classification, we applied filter feature selection before the classification.

Two types of TFCs were considered as noises to be excluded. One was the noise feature of each group. If a feature contained less than 1% of the TFCs of a group, the feature was labeled as random noise from the corresponding group. Another was the outlier TFCs in the remaining features. TFCs corresponding to outliers in any direction of T, f, or E were also identified as random noise. Each group deleted its own noise feature and TFCs. The features composed of the remaining TFCs after removing random noise were used for feature selection.

In order to measure the value of each feature in the classification, we constructed the index G of the feature. Using the filter feature selection method, all features were evaluated as follows:(6)$$G^{j}=\sigma^{2}([\frac{N_{f_{j}=1|C1}}{N_{C1}},\frac{N_{f_{j}=1|C2}}{N_{C2}}\ldots \frac{N_{f_{j}=1|Cm}}{N_{Cm}}]),$$
where $N_{f_{j}=1|Cm}$ represents the number of TFCs of class m at feature *f_j_*, *N_Cm_* represents the total number of TFCs of class m, and *σ*^2^ represents the variance.

The larger the G value of a feature, the greater the difference between classes. Features with the smallest 10% of G were excluded. Each class had a feature selection pattern, denoted as FS_pattern_N, FS_pattern_C, FS_pattern_T, and FS_pattern_L for the normal, cervical spine injury, thoracic spine injury, and lumbar spine injury data, respectively. The SEPs whose TFCs were all noise were excluded. The denoised TFCs were used for subsequent analysis.

### 2.5. Classification of SEP Time-Frequency Components

In this study, naive Bayes was used to distinguish the SEP TFD of SCI at different locations, so as to realize the identification of SCI locations.

Naive Bayes has become one of the most efficient learning algorithms [26]. A naive Bayes is a probabilistic classifier that is based on Bayesian theory with the assumption of attribute conditional independent [27]. It is noted that Bayesian theory is a mathematical formula used to determine the conditional probability of events. The most important step of the classification is to obtain the posterior probability according to Bayes’ theorem:(7)$$P(c|x)=\frac{P(c)P(x|c)}{P(x)},$$
where P(c|x) is the posterior probability, representing the probability that the given sample *x* belongs to class *c*. *P*(*c*) represents the class prior probabilities, P(x|c) is the class-conditional probability of *x* conditioned on class *c*, *P*(*x*) is the prior probability of *x*.

Based on the attribute conditional independence assumption, Equation (7) could be rewritten as follows:(8)$$P(c|x)=\frac{P(c)}{P(x)}\prod_{i=1}^{d}P(x_{i}|c),$$
where *d* is the number of attributes, *x_i_* represent the value of the i-th attribute for the dataset. *P*(*x*) is same for all classes *c*, so the classifier expression can be written as follows:(9)$$h(x)=arg\;\;\underset{c\in C}{\max\;}P(c)\prod_{i=1}^{d}P(x_{i}|c).$$

In this paper, the attributes represent the time-frequency features.

In order to know the accuracy of the method proposed in this paper, we compared the performance of the classifier with that of the support vector machine (SVM) used in a previous study. In the dataset of this study, the classification methods of k-medoids + naive Bayes, k-means + naive Bayes, k-medoids + SVM, k-means + SVM, and SVM were recorded with the highest accuracy. For a specific breakdown of the SVM, please refer to [11].

## 3. Results

### 3.1. SEP Waveforms and Time-Frequency Analysis

Compared with that of the normal group, the SEP of the SCI groups showed decreased amplitude and delayed latency, as shown in Figure 3. Neurophysiologists typically identify SCI by a >10% increase in latency and a 50% decrease in amplitude. According to the SEP waveform in Figure 3, SCI could be diagnosed in all injury groups, but its specific location was difficult to identify. The TFD of the SEP enriched the details of the SEP, from which we could clearly quantify each TFC. The difference in TFD patterns may be a new method for SCI location recognition. According to the TFD in Figure 3, the TFCs with the highest energy showed a decrease in frequency. The distribution of the two components with the highest energy was relatively stable, wherein low-frequency TFCs were thought to be subsequent waves caused by the other strong neural responses. The distribution of low-frequency TFCs of 25–40 ms was little affected by SCI, but its energy was increased close to that of the highest-energy TFCs. Furthermore, lesions involving the dorsal column led to the prolongation or polyphase of small components in the SEPs. Thus, the number of other TFCs with lower energy increased and had higher energy values than the corresponding components of the normal group. The intergroup differences in the distribution of these lower-energy TFCs were more pronounced than those of the higher-energy TFCs.

### 3.2. Clustering of Time-Frequency Components and Feature Selection

In this study, the purpose of TFC clustering was to divide the time-frequency space, and the divided time-frequency region was regarded as the feature. In order to extract the differences in the number of TFCs in different features, the number of clustering centers should be increased as much as possible. However, too many clusters will lead to overfitting. As the results show in Figure 4, when the number of cluster centers ranges from 2 to 150, the silhouette coefficient had a small change, ranging from 0.4 to 0.5. With an increase in the number of cluster centers, it gradually decreased and stabilized at 0.4. We set the number of clusters to 100, which corresponds to the minimum number of clusters as the silhouette stabilizes. We also recorded the curve of classification accuracy changing with the number of clusters. Clustering was randomly repeated 50 times for each set of clusters, selecting a different start point each time. The accuracy showed a trend of increasing and then decreasing as the number of clusters increases. The accuracy reached the maximum when the clustering number was close to 100.

In the process of feature selection, the features labeled as random noise and of low value to the classification were deleted, and the corresponding TFCs of outliers in the remaining features were also deleted. The distribution of the TFCs before and after feature deletion is shown in Figure 5a,b, and the change in the TFD was mainly due to the deletion of low-classification-value features. In the low-value features, the proportion of TFCs in each group was similar, and the total amount was extremely large. Therefore, Figure 5a mainly represents the distribution of these low-value features, while other features were shown as an extremely weak background. When these features were removed, other features with a relatively small number of TFCs but significant differences between groups could be seen, as in Figure 5b. Finally, 20, 19, 24, and 27 features were retained in the normal, cervical, thoracic, and lumbar SCI groups, corresponding to 45.0%, 39.0%, 43.6%, and 60.0% TFCs for each group, respectively. Forty-seven of the features were flagged by all groups to be deleted. If all the TFCs of an SEP waveform are in the 47 features above, there will be no TFC for analysis and therefore it will be deleted. The overall rejection rate of the data was 12.55%. The feature-deletion patterns of each group were recorded separately for training and testing of the classifier.

The TFCs of each group clustered in different regions, and the TFD patterns were related to the location of the SCI. Before feature selection, noise and low-value features resulted in the overlapping color blocks in Figure 5a. These features with small intergroup differences will interfere with the data mining of TFD related to the SCI location. Excluding these features from analysis through feature selection was beneficial for improving the classification accuracy. As shown in Figure 5b, previously neglected features were highlighted, and the intergroup differences were more obvious in these features. Only the features and TFCs in Figure 5b were used for classifier training. The probability density curve in Figure 5 shows that feature selection could highlight the intergroup differences of the TFD and reduce the interference of noise components. These time-frequency regions with the largest distribution differences, such as the latency range of 16–25 ms and 25–30 ms and the frequency bands of 0–50 Hz and 50–120 Hz, were the most sensitive regions to the change of the SCI location.

### 3.3. Classification

Our classifier showed satisfactory classification effects. Table 2 and Table 3 are the confusion matrixes of the classification results. The rows of the matrix correspond to the identification results, and the columns correspond to the true categories. The diagonal elements correspond to correctly classified samples, and the off-diagonal elements correspond to incorrectly classified samples. The percentage of the total number of observed samples is shown in each cell. The recall is listed in the rightmost column to show the level of success in classifying a class. Precision, which is the level of accuracy between information and predictions, is listed in the bottom row. Our method, using the TFD of the SEP, achieved the goal of SCI location identification. Finally, the overall accuracy of the four categories (normal, cervical, thoracic, and lumbar SCIs) was 84.87%. The mean accuracy of cervical-SCI-level (normal, C5, and C6) classification was 88.28%.

By comparing the accuracy of several methods, as shown in Figure 6, we found that the k-medoids + naive Bayes method proposed in this paper achieved the best classification effect. The accuracy of the k-means + naive Bayes method was slightly lower, at 77.34%. The accuracy of the naive-Bayes-based algorithms was higher than that of the SVM. Feature selection based on different clustering methods had little influence on the accuracy of the SVM. The accuracy of k-medoids + SVM and k-means + SVM was 64.13% and 62.09%, respectively. When feature selection was not used, the accuracy of SVM decreased to 42.14%.

### 3.4. Important TFC Distribution Regions

Figure 7 shows the distribution of TFCs after these features were deleted. For any feature, as long as any group did not mark it as noise, all groups will retain this feature. This is because we want to explore the differences in the TFC parameters among different groups in certain time-frequency spaces. Therefore, we did not delete the TFC in this time-frequency space, while in Section 3.2, feature deletion only depended on whether the current group marks it as noise, without considering other groups. According to whether TFC clusters of the normal group exist around the TFC clusters after SCI, the changes in the TFD caused by SCI could be divided into two patterns. For example, in the region of interest (ROI) of R1, R2, and R3 defined in Figure 7, there were TFC clusters from both before and after the injury. In this case, the correspondence of TFC clusters before and after SCI could be established, and the parameters of the TFCs could be statistically tested for each group. While in the ROI of R4 and R5, there was no normal TFC cluster. In this case, the influence of the SCI could be observed from the number of TFCs. Figure 8 plots the latency, frequency, and data proportion of TFCs in each ROI (the proportion of the number of TFCs distributed in the ROI to the total number of current groups). The latency and frequency of TFCs in each group within R1, R2, and R3 were statistically tested.

In R2, the latency of each injury group was significantly longer than that of the normal group. This region was the distribution region of the TFCs with the highest energy. The SCI caused prolonged latency of this peak, which is consistent with the results of previous studies [11,12]. The location changes of SCI also influenced the distribution of TFCs in frequency, but the SCI groups did not show consistent change. In the R4-5 region, the TFCs of the normal group were very few. However, all the SCI groups had TFC clusters in the same region. As shown in Figure 8c, the proportion of data of all SCI groups in R4 and R5 (6.8–10%) was higher than that in the normal group (2.3% and 3.8%). In contrast, in the R3 region, the proportion of data from the SCI groups (14.9–18.2%) was lower than that from the normal group (26.2%). In the region of R4 and R5, the proportion of TFCs in each SCI group was comparable.

## 4. Discussion

SCI is a serious complication of corrective scoliosis surgery and may even lead to paraplegia [1,2]. Direct spinal cord distraction is a common type of SCI injury in scoliosis correction surgery. Timely removal of the SCI source can effectively reduce or even avoid SCI [4,5]. In order to accurately detect the source of distraction SCI during surgery, we developed an SCI location identification algorithm based on k-medoids clustering and naive Bayes, utilizing the correlation between the SCI location and the SEP TFD and achieved satisfactory classification results.

The SEP waveform contains a series of TFCs, which have different origins along the somatosensory pathway [28,29,30]. The latency and amplitude of each component peak can be used to explain the changes in neural activity. Therefore, the SEP can effectively evaluate the physiological integrity of the spinal cord, and it has been widely used as intraoperative electrophysiological monitoring tool for the spinal cord [31]. Animal models of spinal cord compression and contusion have been established, and the SEP has been used to predict the specific location of the SCI. At present, there are relatively few studies on the location prediction of direct spinal cord distraction [5], so we established an animal model of distraction injury. Clamps coupled to a distraction injury were placed on the corresponding spinal segment. Then, the respective clamps were distracted rostrally and caudally to produce a displacement of 3 mm and held for 1 s before being returned to their initial position [20]. In the current study, we used this rat distraction model to evaluate how naive Bayes used the SEP to predict the location of the SCI.

Stable SEP components may correspond to unique anatomical structures of the somatosensory pathway [13,32]. In traumatic or congenital cases, SEP peaks may be delayed or disappear [33]. Previous studies had used the energy peak of the maximum power in TFD as an indicator of SCI in intraoperative monitoring, while ignoring other components with relatively low energy [34,35]. As reported, there were many stable sub-TFCs outside the main SEP TFC region, and these sub-TFCs were potentially associated with pathological information [17]. Furthermore, there were also common sub-TFD changes in different injury locations, suggesting that this sub-TFD change was likely to be a product of spinal neuropathy [13]. This series of studies showed that smaller TFCs contain useful information about the pathological process, especially the location of the SCI. The results of this study showed that the latency of TFCs distributed in R2 in all SCI groups was significantly longer than that in normal group, and the number of TFCs with latency between 35 and 50 ms was greater than that in the normal group.

A recent study recorded both the SEP and the motor evoked potential (MEP) [20], which monitor sensory tracts and the corticospinal tract, respectively. It was found that both SEP and MEP were influenced by the type of SCI. The study only extracted the latency and amplitude of SEP and MEP and did not establish classifiers. However, this has suggested the significance of the MEP in SCI identification. In the follow-up study, it is justifiable to perform more complex feature extraction for SEP and MEP and to establish SCI location classifiers.

In previous studies, TFCs were classified into three categories according to their energy. The TFCs with the highest energy were called the main component, and the others were called sub-TFCs. The subcomponents with energy more than 20% of the main component were called the middle-energy components, and the subcomponents with energy less than 20% were called the low-energy component. These categories are similar to the definition of features in this study. 

Nevertheless, there are some shortcomings to this feature extraction method. On the one hand, the intergroup correspondence of features was unstable. The main component can be used to detect the occurrence of SCI based on the reduction in the energy of the main component. When the energy of the original main component is greatly reduced or there is strong noise, other TFCs may become the highest-energy component. Therefore, the feature correspondence determined by this method is unstable. The correlation coefficients of latency and frequency help to explore this correspondence. However, it works only in a few regions [36]. Sub-TFCs, which have a wider distribution region, were also helpful to identify the location of the SCI [11,12]. In this paper, the TFCs were clustered, and the clustering model was applicable to all groups. Thus, the intergroup differences of TFCs could be directly compared.

On the other hand, in the face of a complex SCI, the optimal energy threshold for feature extraction of the original method may require frequent and complex adjustments. The time-frequency space region involved in the middle- and low-energy TFCs is obviously wider than that of a single feature in this paper. This will affect the selection of the classification method. When the TFD boundary is used for classification, a relatively complex TFD of features is good to improve the accuracy, as when using the SVM [37]. However, the change in the energy threshold will greatly affect the boundary. When the quantity information of TFCs is used, it is better when the features are concentrated in local areas. This study not only used the quantity but also extracted the TFD information through feature selection. Therefore, the division of the time-frequency space was more detailed.

The comparison of accuracy between different classification methods may validate the above hypothesis. Figure 6 shows that the SVM had lower accuracy than the naive Bayes, especially after feature selection. The use of feature selection distinguished a large number of TFC overlaps in each group. An appropriate energy threshold may also reduce the overlap, but it is difficult when many SCI locations are involved.

The SEP signals collected during the operation may contain power line interference, as well as artifacts of eye movement, EMG, and ECG [38]. These artifacts are outliers in the TFCs. We designed the SCI location identification method with the consideration of suppressing the artifact interference. For the feature extraction method, the k-medoids algorithm can deal with outliers better than the k-means algorithm can [24]. For the classification method, naive Bayes uses probability as the classification basis and needs to use all features during each prediction, and hence, it is relatively insensitive to noise and missing values in the training and test data [39]. Figure 6 shows that the k-medoids + naive Bayes method achieved the highest accuracy.

In the current study, we achieved a consistent time-frequency space division pattern for each group by clustering TFCs. The feature selection method was used to extract the SEP TFD information sensitive to the SCI location. Then, the quantity of TFCs in each feature was used to identify the SCI location (normal, cervical, thoracic, and lumbar; normal, C5, and C6). The results show that our classification method achieved good results. The classification accuracy of the cervical SCI level by this method was similar to that in previous studies [11,12]. At the same time, the average accuracy for normal, cervical, thoracic, and lumbar reached 84.8%. This suggests that the SEP has the potential to localize SCI in other locations outside the cervical spine. In addition, the joint method of k-medoids clustering algorithm and naive Bayes classifier proposed in this study provides a new method for intraoperative SCI localization based on the SEP.

In order to avoid the problem of combination explosion and sparsity problems when solving a Bayesian theorem, naive Bayes introduced the conditional independence hypothesis. This assumption is often difficult to hold in real applications, but naive Bayes can achieve quite good performance in many cases. It is assumed that the naive Bayes classifier can produce correct classification results as long as the conditional probability ranking of each category is correct. It is also assumed that the dependencies between features may offset each other, so even if the dependency is ignored, the naive Bayes classifier can still obtain good classification results. However, the impact of the dependencies between features on current research is still unclear [40,41]. Therefore, some naive Bayes methods that analyze the dependency between features may achieve better results, such as semi-naive Bayes or Bayesian net. Despite the limitations, this study extracts the probability information of TFCs from the TFD and applies a new detection method to the identification of the SCI location. Some vertebral segments are not covered in the current study, and the amount of data for each segment is still limited. A large-scale study is expected to be conducted. With an increase in the amount of data, the SCI location recognition method proposed in this paper should improve. Moreover, the association between SCI locations and TFD patterns can be further verified.

## 5. Conclusions

In this study, we found there is a significant delay in the latency of TFCs distributed between 15 and 30 ms and 50 and 150 Hz in all SCI groups, with the most significant intergroup differences among all SCI groups. The TFCs in this region are likely related to the SCI location. Compared with the SVM-based TFC classification method used in previous studies, it was confirmed that the combined method of k-medoids and naive Bayes has a higher accuracy. SEP TFCs can be used to distinguish SCI in the cervical, thoracic, and lumbar regions, which may provide a new noninvasive method for intraoperative SCI localization. The validity of this technique needs to be further verified in future studies.

## Figures and Tables

**Figure 1 bioengineering-10-00707-f001:**
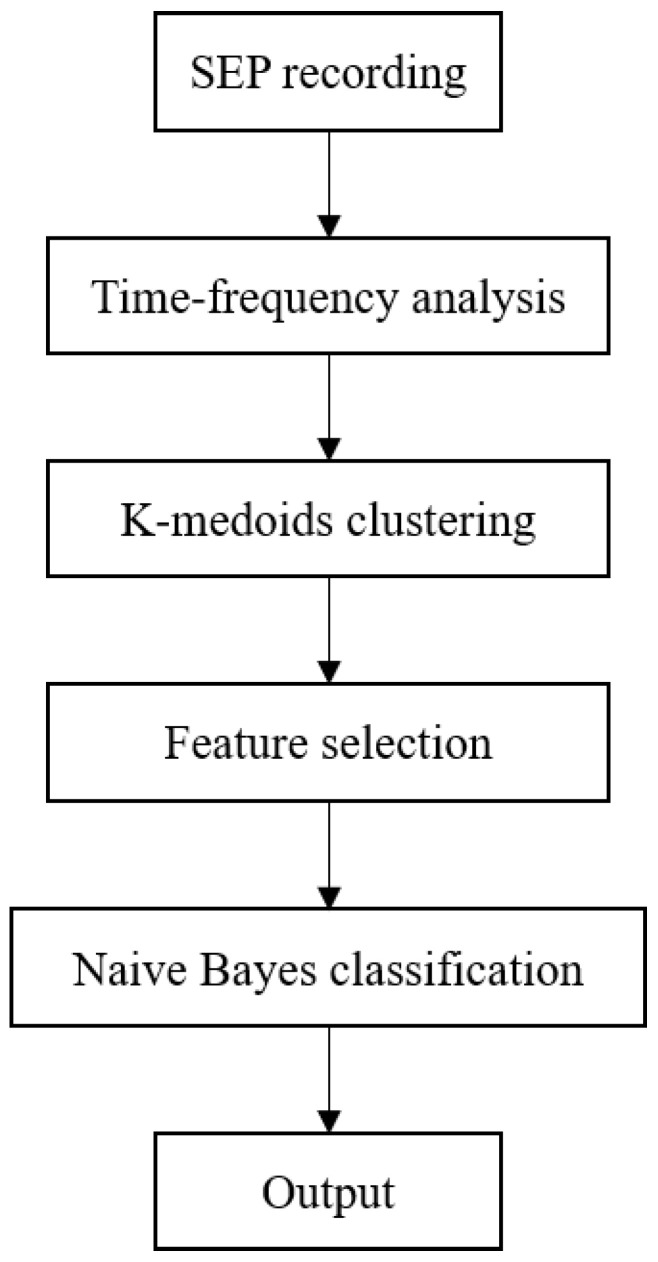
The data processing flow chart.

**Figure 2 bioengineering-10-00707-f002:**
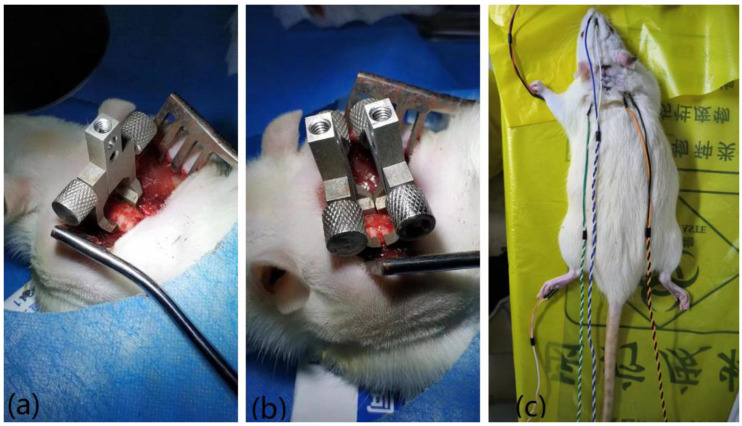
The surgical procedure of SCI rat model. (**a**) Dorsal ligament resection and facet arthrotomy; (**b**) fixation of the vertebral clamps; (**c**) return to initial position and data collection.

**Figure 3 bioengineering-10-00707-f003:**
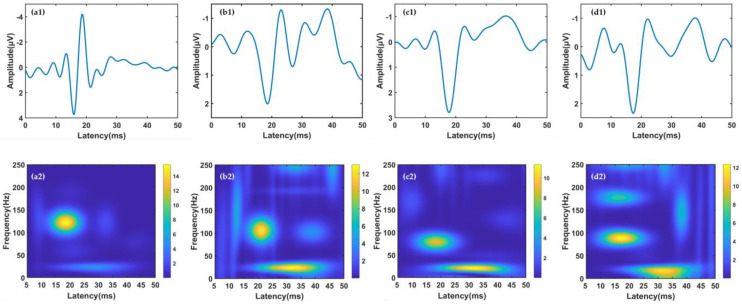
The waveform and MP-based TFD of an example SEP signal. (**a1**,**a2**) The normal group; (**b1**,**b2**) the cervical group; (**c1**,**c2**) the thoracic group; (**d1**,**d2**) the lumbar group.

**Figure 4 bioengineering-10-00707-f004:**
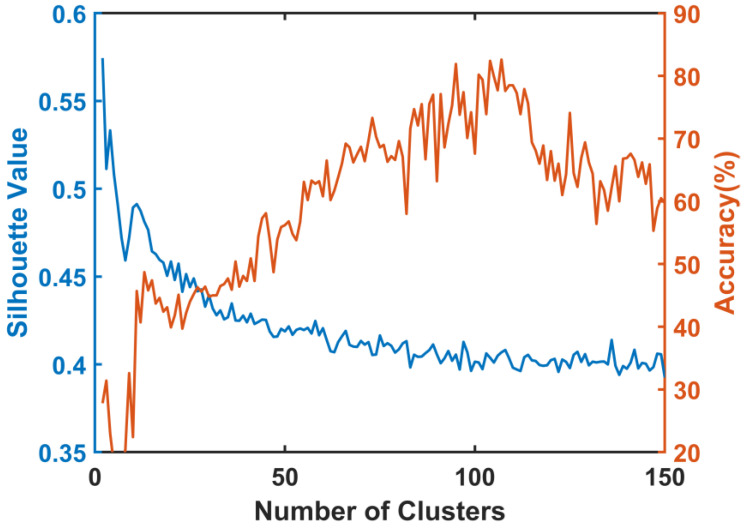
Silhouette coefficient and classification accuracy for different numbers of clustering centers.

**Figure 5 bioengineering-10-00707-f005:**
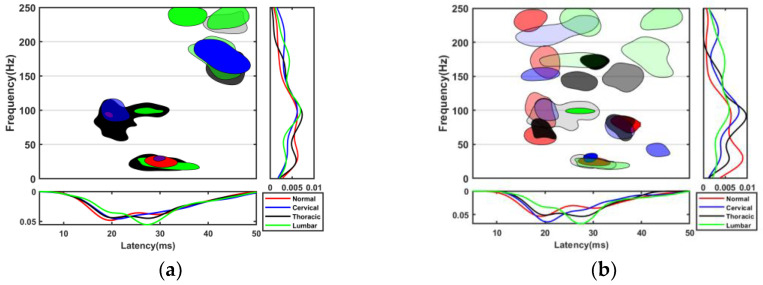
(**a**) TFD before feature selection and the probability density of TFCs in the direction of latency and frequency; (**b**) TFD after feature selection and the probability density of TFCs in the direction of latency and frequency. Red, blue, black, and green correspond to the normal, cervical, thoracic, and lumbar groups, respectively.

**Figure 6 bioengineering-10-00707-f006:**
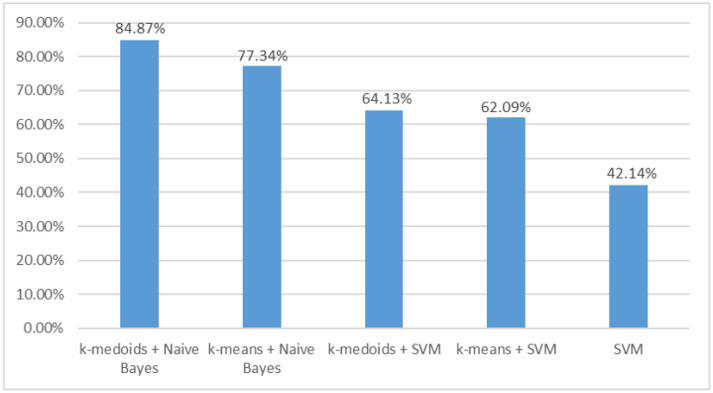
The overall accuracy of different methods. The clustering method of k-medoids and k-means indicates that feature selection was performed.

**Figure 7 bioengineering-10-00707-f007:**
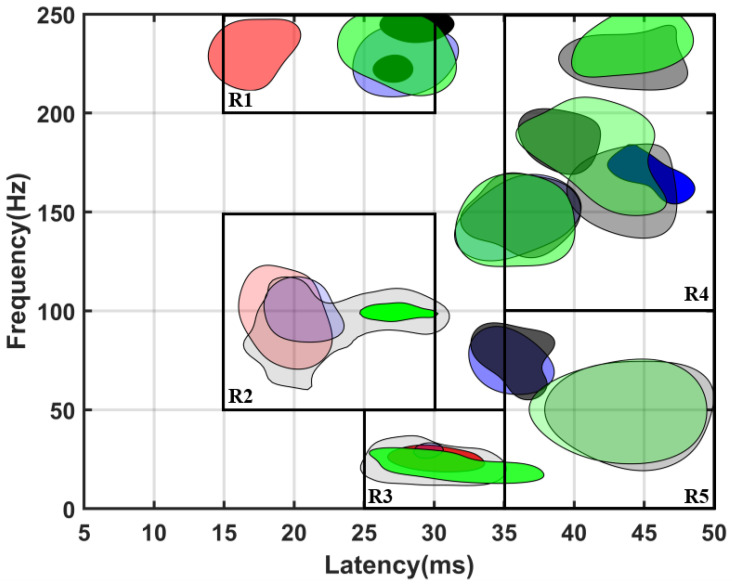
TFD of each group after deleting the common features. Red, blue, black, and green correspond to the normal, cervical, thoracic, and lumbar groups, respectively.

**Figure 8 bioengineering-10-00707-f008:**
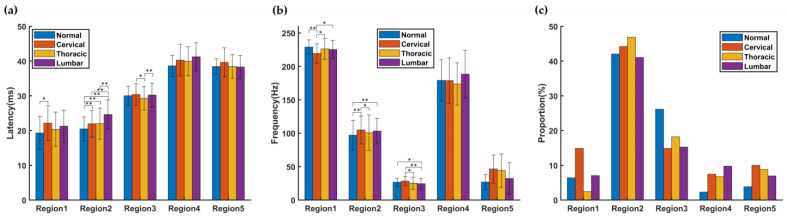
Statistical results of parameters of TFCs in ROI. (**a**) Latency, (**b**) frequency, and (**c**) data proportion of TFCs in each ROI. * *p* < 0.05 (rank sum test), ** *p* < 0.01 (rank sum test).

**Table 1 bioengineering-10-00707-t001:** Experiment grouping design.

Group	SCI Segment	Rats
Normal	−	20 (9.5%)
Cervical	C5, C6	20 (9.5%)
Thoracic	T1–T4 and T7–T13	110 (52.4%)
Lumbar	L1–L6	60 (28.6%)

**Table 2 bioengineering-10-00707-t002:** Classification results for normal, cervical, thoracic, and lumbar groups.

			Predicted				Recall
			Normal	Injury			
				Cervical	Thoracic	Lumbar	
Actual	Normal		8.92%	2.09%	0.78%	0%	75.66%
	Injury	Cervical	0.70%	9.78%	0.00%	0.54%	88.73%
		Thoracic	2.02%	0.00%	37.08%	0.39%	93.91%
		Lumbar	1.16%	1.86%	5.59%	29.09%	77.16%
Precision			69.70%	71.19%	85.36%	96.90%	84.87%

**Table 3 bioengineering-10-00707-t003:** Classification results for normal, C5, and C6 groups.

			Predicted			Recall
			Normal	Injury		
				C5	C6	
Actual	Normal		41.41%	0.78%	3.15%	91.38%
	Injury	C5	3.13%	26.04%	2.34%	82.64%
		C6	1.30%	1.04%	20.83%	89.89%
Precision			90.34%	93.46%	79.21%	88.28%

## Data Availability

Not applicable.

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
