# Peer review of "Identifying Intraoperative Spinal Cord Injury Location from Somatosensory Evoked Potentials’ Time-Frequency Components"

_bioengineering, 2023, doi:10.3390/bioengineering10060707_

Round 1

Reviewer 1 Report

In this study, 19 rat models with different levels of stretch spinal cord injury were established. Somatosensory evoked potentials of hind limbs were collected and 19 time-frequency components of them were extracted. The authors then used k-medoid clustering and Naive Bayes to classify spinal cord injuries at the C5 and C6 levels and those at the cervical, thoracic, and 21 lumbar vertebrae, respectively. The results showed that the latency of 22 time-frequency components distributed between 15-30ms and 50-150Hz was significantly delayed in all 23 groups with spinal cord injury. The overall classification accuracy rates were 88.28% and 84.87%, respectively. The results showed that 24 k-medoid clustering and Naive Bayes methods were able to extract time-frequency component information dependent on the location of spinal cord injury, and indicated that somosensory 26 evoked potentials have the potential to diagnose spinal cord injury. This is an interesting research paper. There are some suggestions for revision.

1.     The motivation is not clear. Please specify the importance of this paper.

2.     Please highlight the contributions of this paper in introduction.

3.     More recently published solutions, especially the solutions published in 2023 and 2022, should be discussed.

4.     To emphasize the difference in the number of different SCI in each TFCs cluster, we increase the number of centroid, that is, dividing the time-frequency space into smaller subspaces. Why divide the time-frequency space into smaller subspaces? What information of TFCs is used by the naive Bayes classifier to identify SCI locations in this partition process?

5.     Why is it said that obtaining reliable TFCS in time-frequency analysis is the basis for subsequent feature extraction and classification? How is TFC represented as a linear combination by MP algorithm? How to calculate the relative energy, frequency, amplitude, span and phase of TFCS?

6.     In the clustering of time-frequency components, why did k-medoids clustering method be used to divide all rat hind limbs SEP TFCs into multi-component clusters? Which object in the cluster is chosen as the center by the k-medoids algorithm? And how to evaluate the effect of the number of clustering centers on the clustering results?

7.     In feature selection, which two steps are involved in random noise removal? How to deal with the features whose number is less than the threshold value? What about the outliers in the remaining features? What do the symbols in formula (6) represent? What does the G value of the feature mean? Why exclude characteristic mixtures containing at least 10% G?

8.     What is naive Bayes used to identify in this study? And what theory is naive Bayes based on? What do formulas (7) and (8) represent respectively? What do the symbols stand for?

9.     More comparative experiments are needed.

English polishing is necessary. 

Reviewer 2 Report

The validity of the approach is somewhat limited; however, naive Bayesian classifier is the way forward in the applicable field. Perhaps, a limitation of the study is the lack of comparison with other classification algorithms such as various clustering approach. It could pave the way to ground your credibility in the approach this study uses and inspire the future researchers to adopt the naïve Bayesian classifier. Overfitting through too many clusters is discussed ok, however, choice of minimum 100 clusters need to be backed by previous studies as well as such a reference could be sufficed.

It has not been mentioned any uses of tools to develop the classifiers. It needs to be well documented. A GitHub link might serve the purpose. The literature review section need to be improved. Having said that it is also important to pinpoint the research questions in a separate paragraph. It is only then the purpose will be well grounded and justified. In line 326, SVM needs to be referenced.

Finally, the future directions of this research need to be improved. Along this line, it needs to be clarified how large-scale studies can verify the findings of this study.

NA

Reviewer 3 Report

The article is interesting and well organised, especially in the initial part of the abstract and introduction. However, I think the following aspects should be considered for clarification and improvement.

In section 2.1, a figure with images of the procedure performed would be useful. In addition to a table with the different groups and percentages.

The sentence "Signals were averaged over 200 responses and filtered between 30 and 3000 Hz". I think it should be explained in a little more detail, justifying the choice of parameters used.

In the results section, I think that the interesting results shown in figure 2 could be explained in a little more detail, explaining what can be seen in the figure according to the authors.

The feature selection process is not explained, only some conclusions of this process are indicated in the paragraph on line 219, which are also confusing.

In the case of figure 4, I think that a more exhaustive explanation of what each shape and colour corresponds to and why, after feature selection, frequency components appear at other latencies, which in the initial TFD seem to be mixed. Perhaps relating it to figure 5 and 6c.

The conclusions section should be expanded by including as a conclusion the comparative and numerical result obtained, as is done in the abstract.

Minor issues

Naive Bayes classifier sounds strange, it is more usual to simply use Bayes classifier. In any case, the word "Naive" should always be capitalised and appears in lower case on several occasions (lines 95, 101, 186, 290, 352 for example).

In line 45, starting with the number "7", I understand that there are too many spaces.

In line 59, there are too many spaces before "However, ...".

The content of the figures should be centred with the page.

In section 2.2, the "MP algorithm" is mentioned, but the acronym "MP" is missing.

In formula number 2, the letter "F" is used for frequency, when it is more usual to use "f" in lower case.

Round 2

Reviewer 1 Report

All my concerns have been addressed. I recommend this paper for publication. 

NA

Author Response

Thank for your review and comments.

Reviewer 2 Report

NA

Author Response

Thank for your review and comments.

Reviewer 3 Report

The authors have taken into consideration many of the indications of the revision. I believe that the text is now clearer.

However, there are some issues that I think need to be better clarified:

In formula number 2 the letter "F" has been changed to "f" for frequency, but that same change should still be made in the text for consistency.

I still think that the results section could be improved by giving more detail of the results shown in figure 3. Explaining what is seen in the figure and not just showing the figure.

The process by which the selection of features has been made remains unclear. How does one go from figure 5a to 5b? How do these figures relate to figure 7?

The conclusion section is still somewhat terse, even though it has been expanded with numerical data, I think it should be expanded a bit more and really serve as an overall conclusion of the study conducted.

Author Response

Thank you very much for your attention and the reviewer’s evaluation and comments on our paper bioengineering-2394374. We have revised the manuscript according to reviewer’s detailed suggestions. Attached file is the responses to your comments point by point.

Round 3

Reviewer 3 Report

I thank the authors for considering the new revisions, improving and clarifying doubtful aspects.